# Living to the High Extreme: Unraveling the Composition, Structure, and Functional Insights of Bacterial Communities Thriving in the Arsenic-Rich Salar de Huasco Altiplanic Ecosystem

Juan Castro-Severyn,[a] Coral Pardo-Esté,[a,b] Katterinne N. Mendez,[c] Jonathan Fortt,[a] Sebastian Marquez,[c] Franck Molina,[d] Eduardo Castro-Nallar,[c] Francisco Remonsellez,[a,e] Claudia P. Saavedra[b]

[a]Laboratorio de Microbiología Aplicada y Extremófilos, Facultad de Ingeniería y Ciencias Geológicas, Universidad Católica del Norte, Antofagasta, Chile
[b]Laboratorio de Microbiología Molecular, Facultad de Ciencias de la Vida, Universidad Andres Bello, Santiago, Chile
[c]Center for Bioinformatics and Integrative Biology, Facultad de Ciencias de la Vida, Universidad Andres Bello, Santiago, Chile
[d]Sys2Diag, UMR9005 CNRS ALCEDIAG, Montpellier, France
[e]Centro de Investigación Tecnológica del Agua en el Desierto-CEITSAZA, Universidad Católica del Norte, Antofagasta, Chile

Juan Castro-Severyn and Coral Pardo-Esté contributed equally to this work. Author order was determined both by contribution and in order of increasing seniority.

**ABSTRACT** Microbial communities inhabiting extreme environments such as Salar de Huasco (SH) in northern Chile are adapted to thrive while exposed to several abiotic pressures and the presence of toxic elements such as arsenic (As). Hence, we aimed to uncover the role of As in shaping bacterial composition, structure, and functional potential in five different sites in this altiplanic wetland using a shotgun metagenomic approach. The sites exhibit wide gradients of As (9 to 321 mg/kg), and our results showed highly diverse communities and a clear dominance exerted by the *Proteobacteria* and *Bacteroidetes* phyla. Functional potential analyses show broadly convergent patterns, contrasting with their great taxonomic variability. As-related metabolism, as well as other functional categories such as those related to the $CH_4$ and S cycles, differs among the five communities. Particularly, we found that the distribution and abundance of As-related genes increase as the As concentration rises. Approximately 75% of the detected genes for As metabolism belong to expulsion mechanisms; *arsJ* and *arsP* pumps are related to sites with higher As concentrations and are present almost exclusively in *Proteobacteria*. Furthermore, taxonomic diversity and functional potential are reflected in the 12 reconstructed high-quality metagenome assembled genomes (MAGs) belonging to the *Bacteroidetes* (5), *Proteobacteria* (5), *Cyanobacteria* (1), and *Gemmatimonadetes* (1) phyla. We conclude that SH microbial communities are diverse and possess a broad genetic repertoire to thrive under extreme conditions, including increasing concentrations of highly toxic As. Finally, this environment represents a reservoir of unknown and undescribed microorganisms, with great metabolic versatility, which needs further study.

**IMPORTANCE** As microbial communities inhabiting extreme environments are fundamental for maintaining ecosystems, many studies concerning composition, functionality, and interactions have been carried out. However, much is still unknown. Here, we sampled microbial communities in the Salar de Huasco, an extreme environment subjected to several abiotic stresses (high UV radiation, salinity and arsenic; low pressure and temperatures). We found that although microbes are taxonomically diverse, functional potential seems to have an important degree of convergence, suggesting high levels of adaptation. Particularly, arsenic metabolism showed differences associated with increasing concentrations of the metalloid throughout the area, and it effectively exerts a significant pressure over these organisms. Thus, the significance

Address correspondence to Francisco Remonsellez, fremonse@ucn.cl, or Claudia P. Saavedra, csaavedra@unab.cl.

Poly-extremophilic bacterial communities from Chilean Altiplano are taxonomically and functionally adapted to thrive under high arsenic concentrations

of this research is that we describe highly specialized communities thriving in little-explored environments subjected to several pressures, considered analogous of early Earth and other planets, that have the potential for unraveling technologies to face the repercussions of climate change in many areas of interest.

**KEYWORDS** extremophiles, metagenomics, arsenic, MAGs, altiplano

Extreme environments such as high-altitude wetlands select for adaptations in bacterial communities that enable them to thrive. This particular and fragile environment resembles life before the oxygenation of Earth and could serve as a model for studying life on other planets. Moreover, microbial communities are critical for maintaining biogeochemical cycles, particularly in extreme environments where there is little presence of other life forms. The Salar de Huasco (SH), a high-altitude wetland located in the Chilean Altiplano (20°18′18′′S, 68°50′22′′W, Chile) at 3,800 masl (meters above sea level), is a Ramsar protected site, considered a hot spot for microbial life (1, 2). This area is labeled as extreme due to the very particular confluence of physicochemical and environmental conditions, including negative water balance, high daily temperature variations, very arid conditions, high salinity, low atmospheric pressure, high solar radiation, and the presence of arsenic (As), among other stressors (3–8).

In high-altitude wetlands, As concentrations, moisture availability, and salt concentrations shape communities at a small scale (9–12). Particularly in northern Chile, the relationship between volcanic activities and the presence of As is a known feature, and geological processes have been attributed to hydrothermal conditions such as geysers and fumaroles throughout the Altiplano (4). Consequently, bacteria inhabiting these environments should exhibit different strategies and harbor genetic machinery to cope with As toxicity, namely, expulsion from the cell (ArsB, Acr3, ArsP, ArsK, and ArsJ), reduction (ArsC), methylation (ArsM), oxidation (AioAB, AoxAB, and ArsH), and respiration or dissimilatory reduction (ArrAB and ArxAB). Frequently, expulsion is commonly coupled with As(III) methylation or As(V) reduction (13–15). Although As-related genes are widespread, they are not universal. In particular, genes such as *aioA*, *arrA*, *arsM*, and *arxA* are less common in the environment than *acr3*, *arsB*, *arsC*, and *arsD* (16).

The presence of toxic metal(oids), such as As, is a significant selection pressure and thus one of the main drivers of the composition of microbial communities (17). Thus, metagenomic approaches have helped to shed some light on the impact of As over microbial communities. For instance, in stromatolites of Socompa, As resistance is achieved mainly through As(V) reduction and expulsion of As(III) via Acr3 efflux pumps (18, 19). Other studies have determined that the presence of metal(oids) can influence biogeochemical cycles, namely, those of C, N, and S, by promoting specific chemical reactions and the enrichment of chemoautotrophs. For example, As(III)-oxidizing bacteria can couple this process with nitrate reduction (20–22), hence highlighting the importance of assessing the metabolic potential of indigenous microbes and communities.

In this study, we tested different established sites in the SH regarding their taxonomic and functional heterogeneity. We focused on As-related genetic elements as well as other relevant metabolic functions. Increasingly, such analyses enable the identification of relatively small subsets of markers associated with a particular ecologically important function. Additionally, their identification and the dynamics of bacterial communities are potentially useful bioindicators for monitoring ecosystem health (23, 24). Therefore, considering the aforementioned parameters, we aim to describe and characterize the composition, structure, and functional potential of bacterial communities from the sediments of five different SH sites along an As gradient.

## RESULTS

The Salar de Huasco possesses an important level of variation within a relatively small area (the maximum separation of sampling sites is 5.9 km), including daily oscillations in a wide range of environmental parameters (temperature and humidity) and

others which vary in spatial gradients, i.e., salinity and As (25, 26) (Fig. 1); see also Table S1 in the supplemental material. In this context, shotgun metagenomic sequencing of the sediment samples representing the five SH sites yielded an average of 87.2 million reads (150-bp length) per sample, with a quality score of ≥30 presented by ~95% of the reads obtained.

**The bacterial communities of SH are highly heterogeneous and rich in unknown taxa.** The substantial differences between the SH sampling sites are reflected by the taxonomic compositions of bacterial communities, which showed contrasting patterns and variation between sites (Fig. 2). Overall, our results indicate that the *Proteobacteria* and *Bacteroidetes* are the most prevalent phyla in the sampled sediments, which are particularly enriched in the H3, H4, and H5 communities, accounting together for >60% of all observed taxa (Fig. 2A). In turn, these phyla represent ~50% in the H0 and H1 communities. Interestingly, the H0 site is dominated by the *Cyanobacteria* phylum (34% of the total community); it also has the highest abundance of *Firmicutes*, *Patescibacteria*, and *Spirochaetes* phyla as well as significantly lower proportions of *Actinobacteria*. Moreover, the H1 community profile presents the highest abundance of *Chloroflexi*, *Actinobacteria*, *Verrucomicrobia*, *Planctomycetes*, and *Acidobacteria* phyla. On the other hand, H3, H4, and H5 communities are more similar to each other, despite the vast differences in low-abundance taxa, many of which are exclusively present in only one community (see Table S2).

The most abundant bacteria at the lowest (available) taxonomic rank reflect the same pattern, with a great abundance of the *Proteobacteria* genera *Roseovarius* and *Desulfotignum* in the H3, H4, and H5 communities (Fig. 2B; see also Table S3). Moreover, *Halomonas*, *Thiobacillus*, *Luteolibacter*, and *Truepera* genera are more widespread, while *Rhodohalobacter*, *Marinobacter*, *Psychroflexus*, and *Brumimicrobium* are also concentrated in H3, H4, and H5. On the other hand, H0 and H1 are enriched in *Cyanobacteria*, such as *Arthrospira* and three members of the *Chloroplast* order, as well as genera with various metabolism types such as *Methylibium*, *Hydrogenophaga*, and *Desulfomicrobium*. In addition, of note is that of the total 3,801 bacterial amplicon sequence variants (ASVs) detected, many were not classified within any known phylum (6.5%) or genus (58.3%). Nevertheless, *Halomonas* and *Marinobacter* genera were detected, both of which are culturable and seem to be recurrent in the SH (27). Of the five communities, that of H4 is by far the most diverse, according to the Observed, Shannon, Chao, and Simpson diversity indices, which were heterogeneous between samples (see Fig. S1). Specifically, the phylogenetic diversity showed that the H0 community has a higher number of distant taxa than the other communities under study. Also, co-occurrence analysis allowed us to infer possible interactions between the communities. The network was composed of 112 ASVs (having at least one significant correlation), a total of 155 interactions (115 positive and 40 negatives), low modularity, and no detectable "keystone" taxa. Spatially, there is a main network comprising most ASVs, where *Proteobacteria* and *Bacteroidetes* phylum nodes have the highest degree of interaction (see Fig. S2). Also, most of the highly connected nodes in the network belong to unknown or undescribed taxa.

As a whole, the beta diversity analysis shows that the dispersion patterns between the communities correlate with the composition analysis. Hence, the taxonomic compositions of H3, H4, and H5 metagenomes are more similar between each other and distinctive from those of H0 and H1, producing three very well-defined groups (Fig. 2C). This contrasts with the obtained alpha diversity indexes, suggesting that their members could differ in abundance. Moreover, the distributions of the most abundant taxa between the five communities could explain this segregation. Furthermore, we investigated to what extent the microbial community structure was explained by the environmental factors and found that many variables exert a significant influence over the structure and distribution/grouping. Of these factors, As appears to be one of the most important driving forces shaping these communities. Additionally, salinity along with As separated the H3, H4, and H5 communities, while H1 segregation appears to be driven by pH. However, H0 segregation was not explained by any of the variables measured.

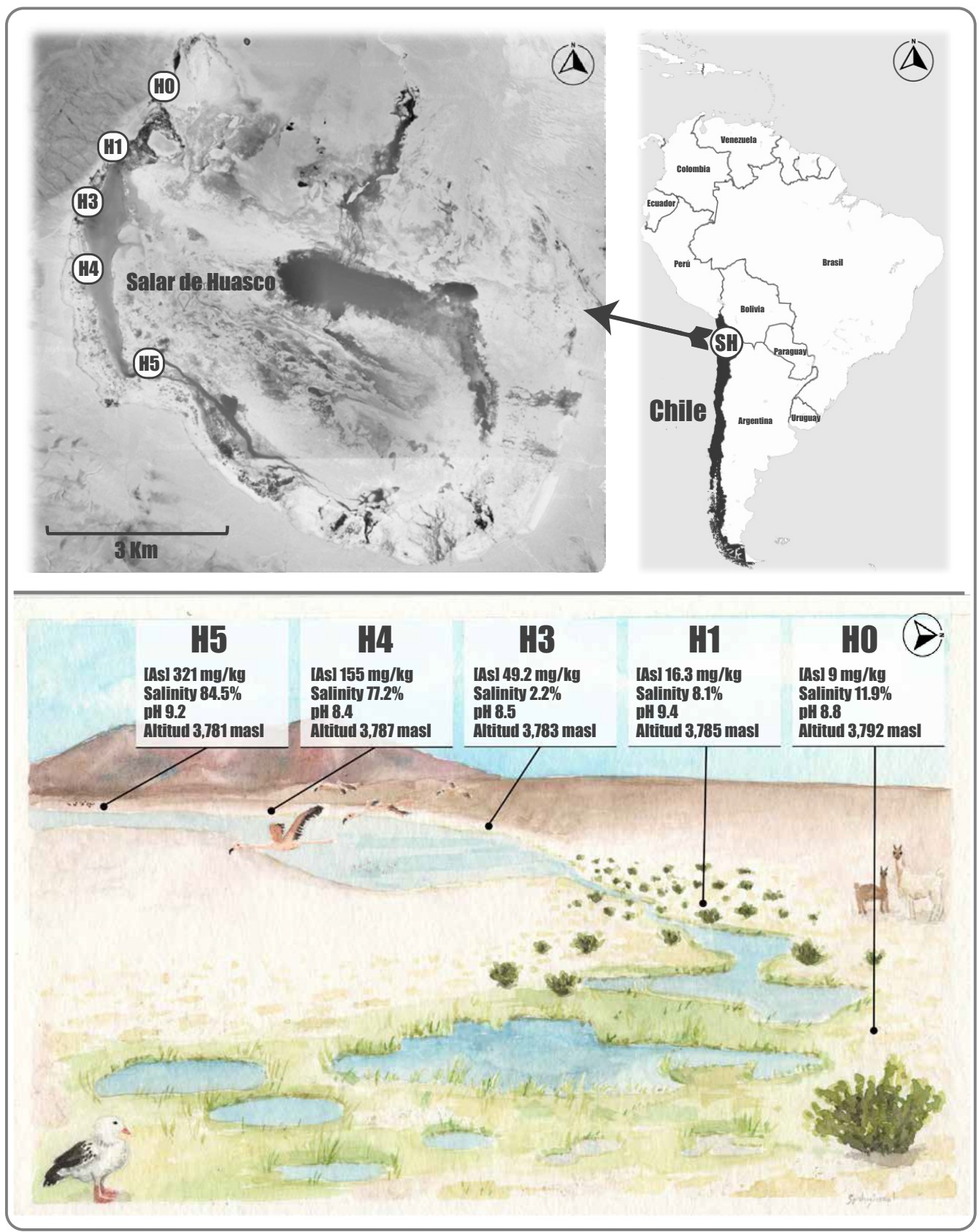

**FIG 1** Salar de Huasco study area. (Top) Map showing the five sampling sites investigated in this study (H0: 20°15′48.8′′S, 68°52′28.4′′W; H1: 20°16′27.7′′S, 68°53′3′′W; H3: 20°16′59.2′′S, 68°53′16.7′′W; H4: 20°17′40.9′′S, 68°53′17.3′′W; and H5: 20°18′37′′S, 68°52′42′′W). The SH is located between 68°47′, 68°54′ W and 20°15′, 20°20′ S in the Tarapacá region of northern Chile (Google Earth). (Bottom) Illustration of the SH landscape seen from the H0 site toward the southwest; the signs show some particular characteristics of each site (according to those reported by Castro-Severyn et al. [8]).

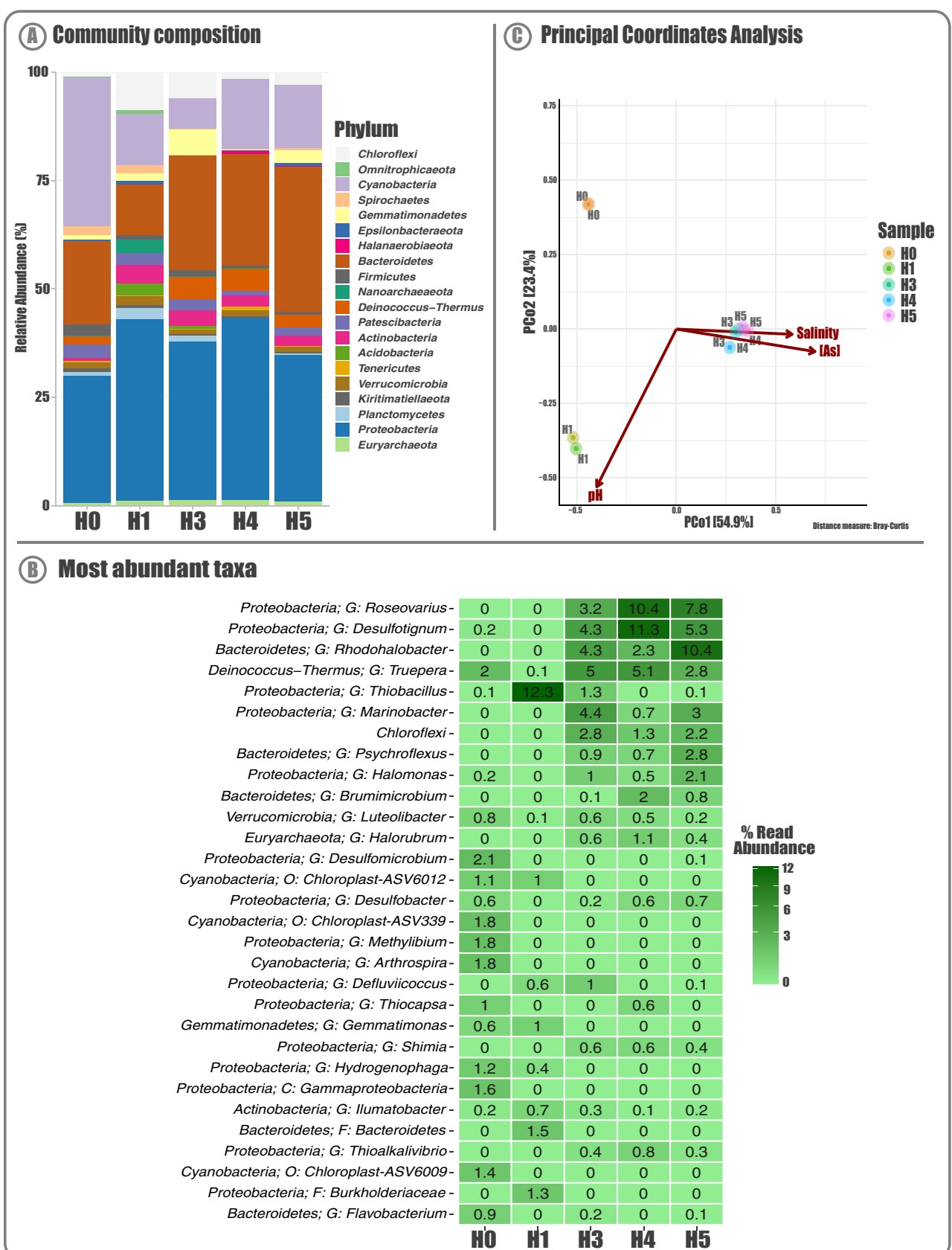

**FIG 2** Salar de Huasco bacterial communities. (A) Taxonomic composition and relative abundance of the SH microbial communities in the five studied sites (H0 to H5); stacked bars show the 20 most abundant bacteria at phylum rank. (B) Composition and structure of the SH bacterial

**Functional approach of SH communities reveals metabolic specialization.** The SH metagenomes we analyzed represent the compositions in surface sediments of 5 different sites with a great variation in As content (from 9 to 321 mg/kg). The 337.5 million quality-controlled reads were coassembled and yielded a total of 994,545 contigs ($\geq$1,000 bp) corresponding to 2.38 million genes, which were used to generate abundance/distributions and functional profiles of the 5 communities under study (Fig. 3; see also Table S4). Of those contigs, 148,394 ($\geq$2,500 bp) were hierarchically clustered and then profiled by read recruitment of the data from the five communities (Fig. 3A). The resulting patterns are highly variable, considering the detection of contigs in each metagenome. In particular, metagenomes from H1 and H5 are the most contrasting, as they harbor 18.78% and 78.40%, respectively, of all available contigs in the SH. The other samples also have diverse percentages of representation (H0, 25.00%; H3, 72.08%; and H4, 48.33%), and their clustering by the read recruitment profiles correlates with the taxonomic profiles presented previously. Specifically, H3, H4, and H5 are more similar to each other, and H0 and H1 are more distanced from each other and from H3/H4/H5. On the other hand, the functional profiles have a higher level of convergence between communities (Fig. 3B). Furthermore, the most enriched categories (SEED subsystems 1) are associated with metabolism (amino acid derivatives, carbohydrates, protein metabolism, DNA metabolism, cofactor, vitamins, prosthetic groups, and pigments), as these are important for bacterial versatility and adaptability to thrive in harsh environments. The stress response, membrane transporter, cell wall, and capsule categories, other features required for prospering under severe conditions, were also enriched.

Nonetheless, statistical testing showed that the differences presented by the metagenomes in most categories were significant (see Table S5). Particularly, the five SH communities showed specific differences in some categories such as As resistance (SEED subsystems 3), following the same tendency previously mentioned, where H0/H1 metagenomes are more similar than H3/H4/H5, with 0.21% to 0.22% and 0.26% to 0.27% of reads recruited by this category, respectively. Other significant categories in the communities were carbon, nitrogen, and phosphate metabolism, zinc, nickel, cobalt, iron, and manganese transport, osmotic stress, the circadian clock in *Cyanobacteria*, and the Calvin Benson cycle. However, when assessing the ability of these bacterial communities to carry out necessary reactions to sustain key biogeochemical cycles, we found that S, N, and $CH_4$ showed significant scores (multigenomic entropy-based score [MEBS] analysis) for all five metagenomes, implying that the necessary metabolic pathways and machinery are present and complete in bacteria isolated from all environments (Fig. S3). We also found representatives of several pathways, including sulfur reduction and oxidation, nitrate fixation and reduction, methanogenesis, and methane oxidation, and the utilization of a wide range of the available nutrients. Particularly, a significant difference is present in the $CH_4$ cycle, which is much more enriched in the H1 and H0 communities; consequently, we only detected methanogens and methane oxidizer taxa in these sites. Conversely, a wide range of sulfur-associated pathways were detected in all sampled sites (assimilation, oxidation, reduction, and utilization of sulfur derivatives), suggesting its importance in environments where volcanic-derived sulfur oxidation products leach and flow into the salt flats (11). Nonetheless, a small decrease in sulfur in H1 was observed.

**Arsenic expulsion is the main mechanism to thrive in the SH.** To gain a broader view on the communities' functional potential related to As metabolism or resistance/tolerance, we determined the abundance of the genes belonging to the known mechanisms (Fig. 4). Again, the abundance pattern of these genes correlates with the

**FIG 2** Legend (Continued)
communities. Heat map showing the 30 most abundant amplicon sequence variants (ASVs; bacterial lineages). Taxonomic determination is shown at the phylum rank, plus the best hit available. C, class; O, order; F, family; G, genus. (C) Beta diversity by principal-coordinate analysis (PCoA) on Hellinger-transformed ASV relative abundances. Each point corresponds to a community from the different sites (represented by colors), and its relative distance indicates the level of similarity to all other samples. The arrows indicate the explanatory power of the statistically significant environmental parameters with regard to the observed variation in community composition.

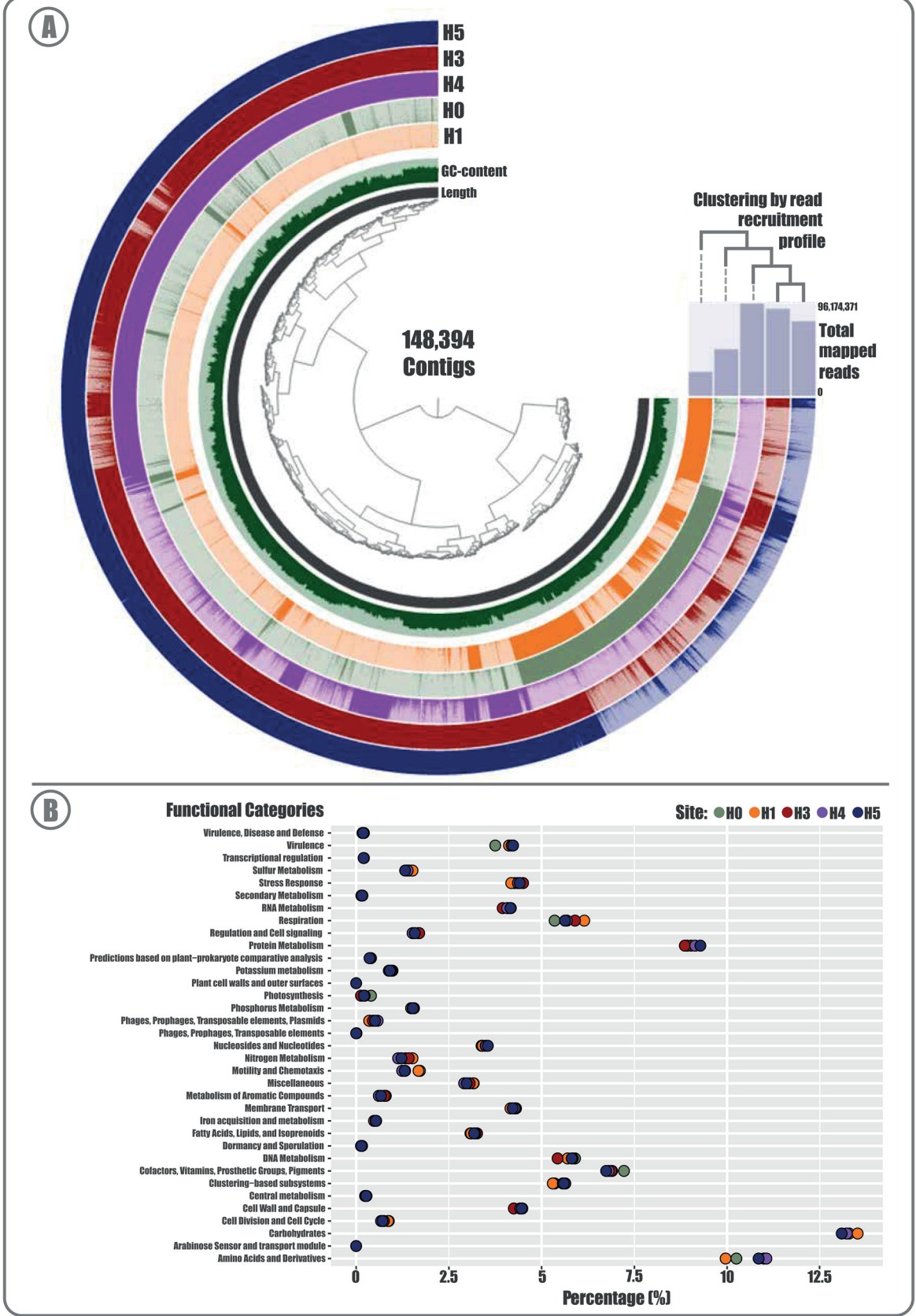

**FIG 3** Salar de Huasco metagenomes. (A) Circular map representing the universe of contigs detected in the five SH sites: the central tree shows the organization of contigs on Ward's linkage with Euclidean distances, whereas the seven circle layers (from the bottom

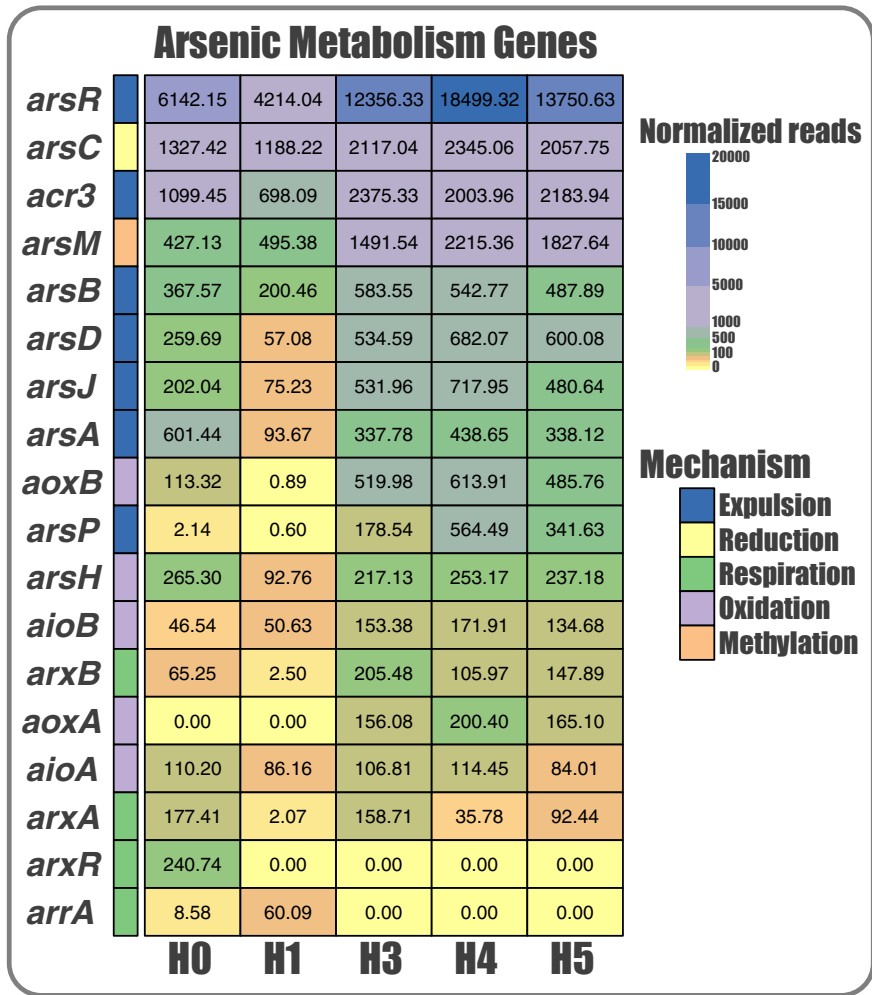

FIG 4 Distribution and abundance of As metabolism genes in the Salar de Huasco. Heat map showing the (normalized) number of reads that aligned against the corresponding protein identified in each sample, according to the color scale. Genes are grouped by colors, representing the 5 As response mechanisms.

observed tendency, as H3, H4, and H5 are more related and are distant from H0 and H1. Moreover, although marker genes of As methylation, reduction, oxidation, respiration, and expulsion mechanisms are present in all sites, most are significantly more abundant in the H3, H4, and H5 sites. This may be due to the much higher As concentration in the sediments of these sites. The *arsR* regulator, the *arsC* reductase and the *acr3* pump were the most overrepresented genes, indicating that the As(V) reduction and subsequent As(III) expulsion strategy is the most prevalent one used by bacteria that inhabit the SH. Moreover, *arsM* was among the most abundant genes, particularly enriched in the H3, H4, and H5 metagenomes. Nevertheless, genes related to oxidation and respiration mechanisms were less abundant yet present an interesting distribution; those related to oxidation (*aoxA*, *aoxB*, and *aioB*) are enriched in H3, H4, and H5, while those related to respiration (*arrA* and *arxR*) are enriched in H0 and H1.

FIG 3 Legend (Continued)

up) represent, for the corresponding contig, its length, GC content, and presence in the five metagenomes. The top right bars represent the number of mapped reads for the corresponding metagenome, and their clustering by read recruitment profile is depicted in the dendrogram. (B) Patterns of functional potential for each metagenome, according to the presence and abundance in the SEED database of metabolic pathways and functions: subsystems at level 1. The circles represent the percentage value for the corresponding category in each metagenome (defined by colors).

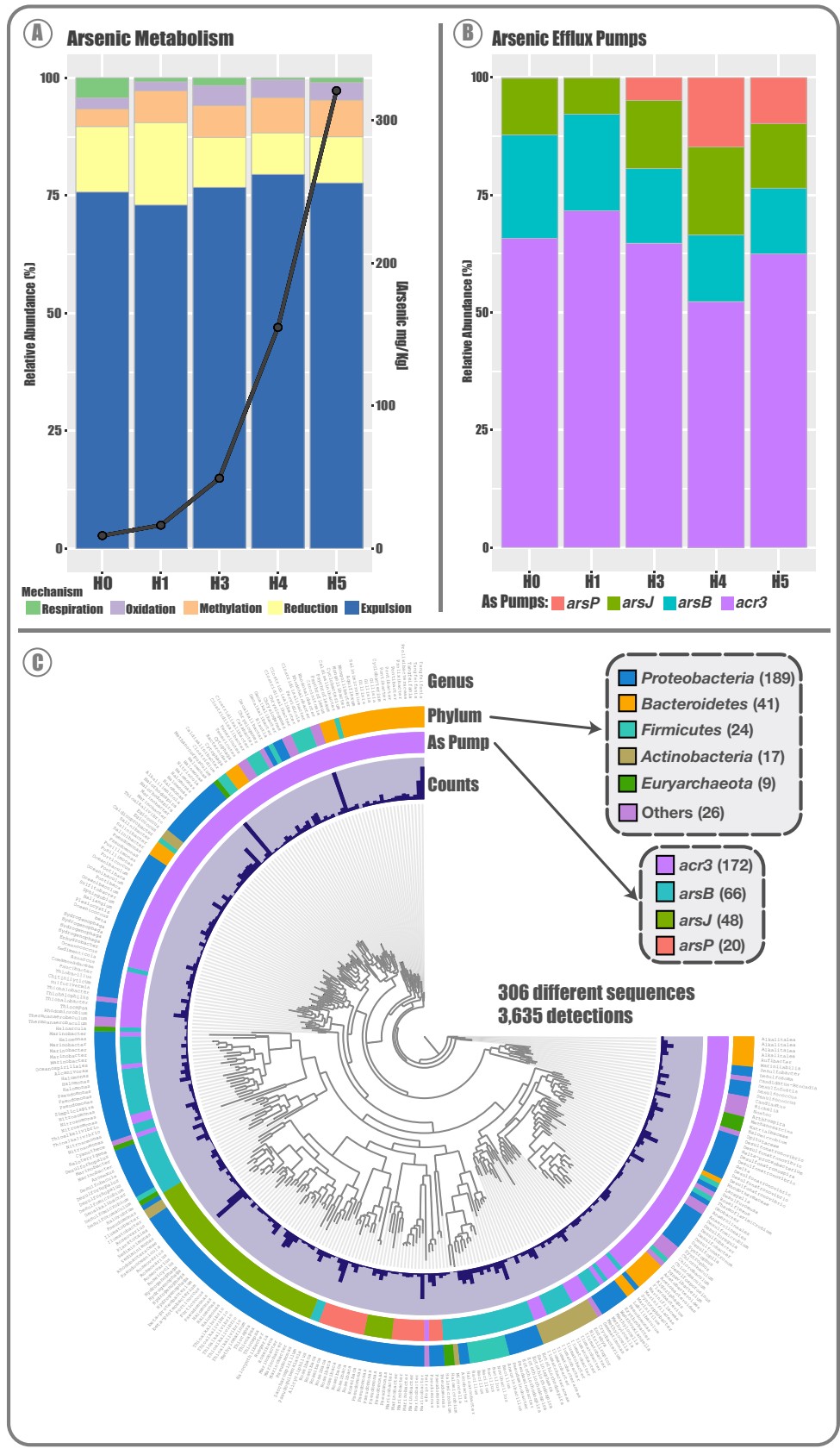

**FIG 5** Arsenic metabolism in the Salar de Huasco. (A) Distribution and abundance of detected genes related to arsenic (As) metabolism in the five SH metagenomes. Stacked bars show the proportion of all genes grouped by

Overall, the proportional distribution patterns of the mechanisms in the 5 metagenomes show some similarities and are quite constant (Fig. 5). Particularly, the expulsion mechanism as a whole was the most abundant one for all sites, covering around 75% of all the sequences (Fig. 5A). The greatest differences were observed in oxidation and respiration mechanisms, as stated above. Additionally, respiration (dissimilatory reduction) and reduction genes are more present in H0 and H1, respectively. Interestingly, the proportion of each mechanism in the five metagenomes is independent of the number of detected sequences and of the As concentration of the site. Furthermore, comparing the abundance of specific As efflux pumps (from the expulsion mechanism), we observe that most sequences correspond to *acr3* (Fig. 5B), followed by *arsB* and *arsJ*. Notably, *arsP*, which is an efflux permease that confers resistance to organic As (as roxarsone and methylarsenite), was more abundant in the sites with higher As concentrations. Moreover, the higher *acr3* variants and abundance could be because this pump is present in a wider number of bacteria phyla, covering most of the diversity observed (Fig. 5C). Also, this efflux pump seems to be more ancestral than *arsP* and *arsJ*, which are only present in *Proteobacteria* and in more recent branches.

**Novel genomes from SH belong to undescribed genera.** The binning process reconstructed 195 bins (which were manually curated and evaluated for completion and redundancy) to gain insights into nonculturable bacteria. This resulted in 19 metagenome-assembled genomes (MAGs) that met the completion of ≥80% and redundancy of ≤10% criteria (Fig. 6); together, they clustered 4.99% of the contigs in the metagenome profile database. The MAGs taxonomic affiliation resulted in 1 belonging to the *Eukarya* domain and 18 remaining in the *Bacteria* domain, distributed in 4 different phyla (9 *Proteobacteria*, 7 *Bacteroidetes*, 1 *Cyanobacteria*, and 1 *Gemmatimonadetes*) (Fig. 6A). We were able to assign 22% (4/18) of the *Bacteria* MAGs to previously reported genera, suggesting a great diversity of novel species that are yet to be described. Also, the contigs numbers of MAGs were very variable, ranging from 208 to 626 among the *Bacteria* (Eukaryota being substantially larger, with 4,174 contigs); the same tendency was observed for the GC content (36.1% to 69.4%). On the other hand, most of the reconstructed MAGS were represented or detected in the H3, H4, and H5 metagenomes, showing differential abundance patterns in each site (Fig. 6B; see also Table S6).

**Novel genomes from SH are mostly arsenic reducers.** The functional potential of reconstructed MAGs was evaluated globally and particularly in relation to their repertoire of As-associated genes (Fig. 7). For these evaluations, only the 12 MAGs that met the quality standards of completion (≥90%) and redundancy/contamination (≤10%) were considered. Therefore, a total of 267 KEGG modules were detected among the MAGs, presenting variable absence/presence and completion patterns (Fig. 7A; see also Table S7). This variation reflects the taxonomic affiliations of the MAGs; all the *Bacteroidetes* (MAG111, MAG12, MAG116, MAG130, and MAG193) clustered together, as well as the *Deltaproteobacteria* (MAG148, MAG143, and MAG93) and *Gammaproteobacteria* (MAG129 and MAG144). On the other hand, the only *Cyanobacteria* (MAG29) is close to the *Gammaproteobacteria* and the only *Gemmatimonadetes* (MAG3) is close to the *Bacteroidetes*. Particularly, MAGs displayed a considerable number of genes regarding As metabolism, where the same tendency shown before was observed, with a dominant presence of genes related to As expulsion (*acr3*, *arsA*, and *arsJ*) throughout all taxa, followed by reduction (*arsC*), methylation (*arsM*), and oxidation (*arsH*) mechanisms (Fig. 7B; see also Table S8). Interestingly, *arsJ* was only detected in one *Proteobacteria*, which agrees with its lower abundance found at the metagenomic level in the SH. Besides, *arsH* was only detected in the *Cyanobacteria*, which also reflects the abun-

**FIG 5** Legend (Continued)
mechanism type based on relative abundance (%), and the line represents total As concentration in each corresponding site (mg/kg of sediment). (B) Stacked bars represent the proportion of all detected As efflux pumps in the five metagenomes, based on relative gene abundance (%). (C) Phylogenetic analysis of 306 nonredundant sequences of As efflux pumps detected in the SH. The layers surrounding the phylogenomic tree indicate detection level, type of efflux pump, taxonomy, the phylum level, and the species.

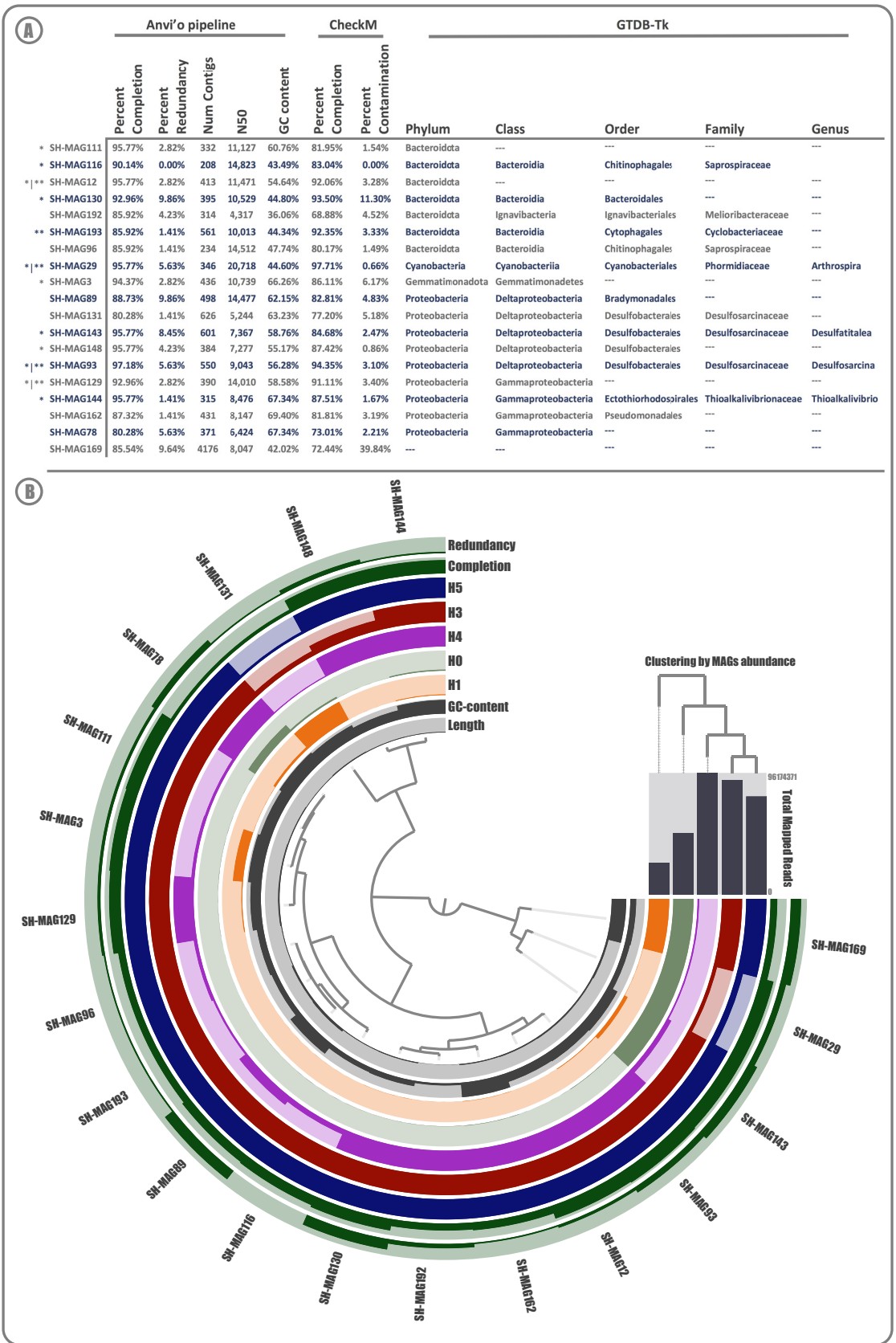

**FIG 6** Salar de Huasco metagenome-assembled genomes. (A) Summary of genomic features of the 19 MAGs in the SH: quality indices and taxonomic affiliation; the results of anvi'o pipeline (*) and/or CheckM (**) meet the quality standard of completion of

dance of this taxa and of this gene globally. Therefore, the presence of some of the genes in the different MAGs is associated with their taxonomy.

## DISCUSSION

There are marked variations in the amount of total As in sediments taken from different points in the Salar de Huasco. We have described this as a gradient from north to south (Fig. 1) (8), which is consistent with the widely reported heterogeneity for this area and the potential impact of such an extreme condition over the resident organisms (29). Previous studies have detected variations in conductivity, organic matter, and dissolved oxygen, even in geographically close areas (5, 6, 30–32). In aquatic ecosystems, the quantity of trace elements and physicochemical states of some elements are related to their circulation/interaction and with the coping mechanisms of the organisms present (33). In particular, salt flats, such as SH, contain more than 50% $CaCO_3$ and high As levels. A fraction of the As lies in sediments as soluble salts ($Na_2HAsO_4$), while it can also be associated with calcium carbonate ($CaHAsO_4$) or adsorbed into Fe and Al oxides (34). The metabolism of inhabiting microbial communities or the activity of primary producers could contribute toward the diversity of the physicochemical properties (35). It is hypothesized that this spatial and functional heterogeneity is the cornerstone of resilience in such extreme ecosystems. Therefore, we must consider the presence of high concentrations of As as a selective pressure, which could modulate the composition of the H3, H4, and H5 communities (grouping them together and separating them from the rest).

The five communities described from the SH metagenomes are dominated by *Proteobacteria*, *Bacteroidetes*, and *Cyanobacteria* (Fig. 2), similar to previous reports for other altiplanic environments (Socompa volcanic lake in Argentina and La Brava Lagoon, Salar de Atacama, in Chile) mostly carried out by 16S rRNA amplicon sequencing (36, 37). Specifically, the same pattern has been reported previously for the SH (2, 31). However, the proportions are different, probably because the shotgun metagenomic sequencing used in this study is more sensitive, which is evidenced by the magnitude of the alpha diversity indexes. Nevertheless, these values are also higher than those reported by other metagenomic studies from similar areas (38).

Moreover, there is evidence of *Proteobacteria* enrichment in places with different As concentrations (11), as many microorganisms described in this group can interact with and tolerate As. Some examples are *Acidithiobacillus* and *Desulfovibrio* genera, which can solubilize As from solid compounds or precipitate it by coupling As and sulfur reduction, respectively (9, 39). The phylum *Bacteroidetes* is widespread and commonly found in hypersaline wetlands and microbial mats (37, 40–44). Furthermore, *Cyanobacteria* play fundamental roles in any community they belong to, as primary producers and participating in bioweathering and matrix transformation processes (45). Also, they are usually abundant in places exposed to sunlight (46). Besides, previous reports have proposed that the cyanobacterial communities in SH are unique (5). In addition, many recurrent representatives of these communities (*Marinobacter* and *Halomonas*) have been cultured and isolated under laboratory conditions (27). Specifically, we isolated *Exiguobacterium* strains from these communities to describe the mechanisms responsible for their high resistance to As (8, 47, 48).

On the other hand, a high percentage of unidentified genera has also been reported in the Chilean Atacama Desert. Indeed, up to 66% of putatively novel taxa belong to this so-called bacterial "dark matter" (49). This is a direct consequence of database shortages, reinforcing the need to keep exploring these unique environ-

**FIG 6** Legend (Continued)
≥90% and redundancy/contamination of ≤10%, according to Bowers et al. (28). (B) Presence and abundance of the 19 MAGs in the five SH metagenomes: the central tree shows the organization of MAGs on Ward's linkage with Euclidean distances, whereas the seven circle layers (from the bottom up) represent, for the corresponding MAG, its length, GC content, and abundance in the five metagenomes. The top right bars represent the total number of mapped reads for the corresponding metagenome, and the dendrogram shows their clustering by absence/presence profile.

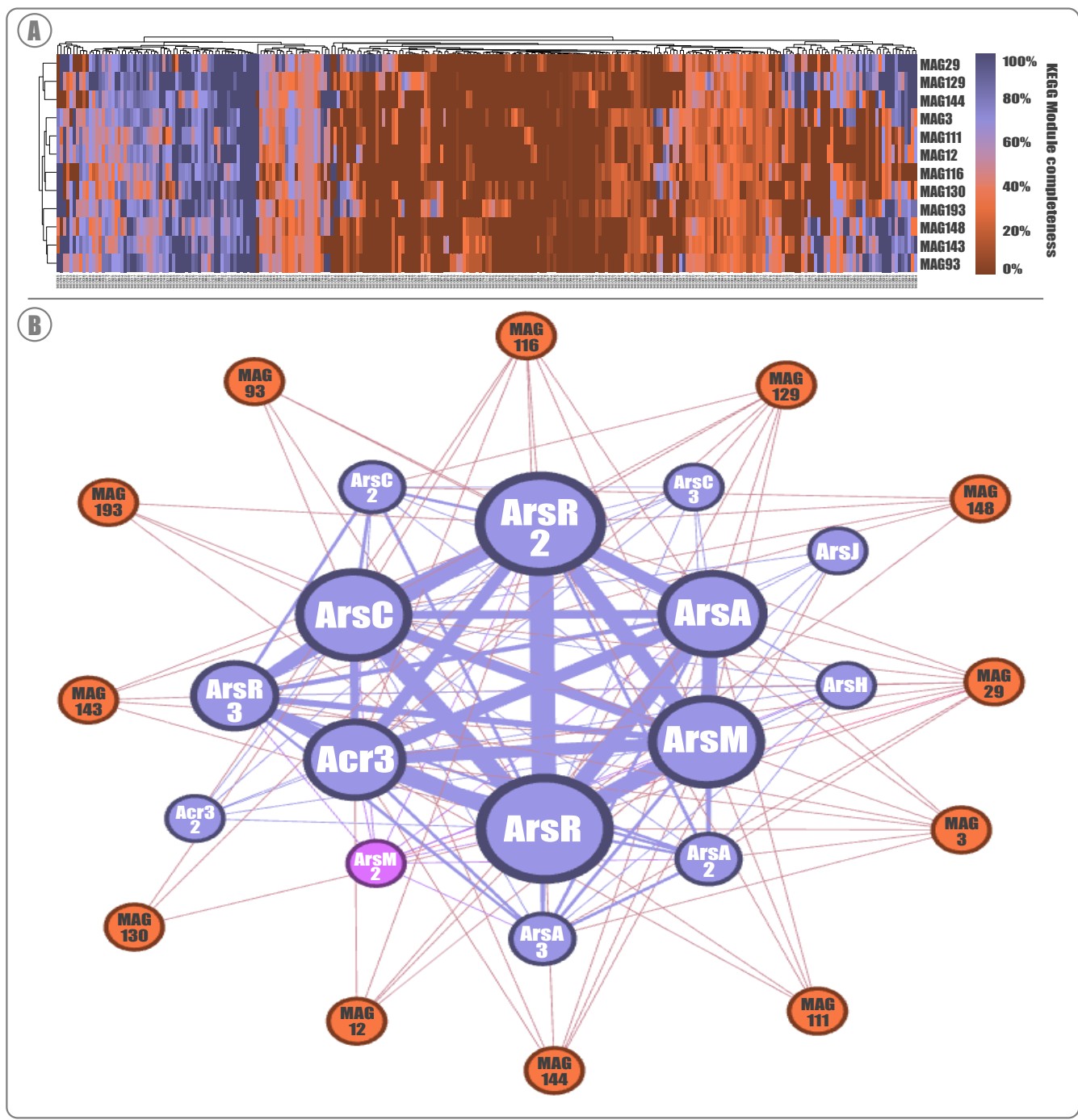

**FIG 7** Functional potential profiles of MAGs: global and As specific. (A) Metabolic capabilities of the 12 selected MAGs according to KOfam (KEGG Orthologs database); the heat map shows the KEGG Modules completion displayed by the color gradient for a given module in each corresponding MAG. (B) Functional network analysis of the 12 MAGs according to the presence and abundance of detected As metabolism genes. The size of gene nodes represents the level of detection, and the size of the edges represents the correlation level. Blue nodes represent genes, and orange nodes represent MAGs.

ments and describe the novel highly adapted microorganisms that remain unknown to science and that are important to these ecosystems. The co-occurrence networks show overlapping taxa among the communities, mainly of unknown and little-studied taxa. Also, we can infer that this is a very fragile and susceptible ecosystem, as no keystone nodes have been detected (50).

At the lowest available taxonomic ranks, we found some bacteria that are typical of extreme environments, such as *Thiobacillus*, a well-known sulfur-oxidizing bacterium,

and *Desulfotignum*, an anaerobic group known to be sulfate reductors, as well as other less-studied groups. The most abundant belong to the *Roseovarius* genus, an aerobic, nonphotoautotroph bacterium. Of this genus, *Rhodohalobacter* is a facultative anaerobic and moderately halophilic group, and *Truepera* comprises aerobic chemo-organotropic alkaliphilic bacteria. This is evidence of the diversity of lifestyles that these microorganisms have, suggesting a cooperative and specialized community. Likewise, this is in coherence with our findings that photosynthetic organisms are enriched in H0 and H1, and *Proteobacteria* are the most abundant group in all the other sampled sites. Other similarities are apparent, as the genus *Methylibium* was only found in H0, where the methane-related metabolism function was found to be enriched in the MEBS analyses. In addition, the importance of some of these abundant taxa is highlighted, namely, *Desulfomicrobium*, *Thiobacillus*, and *Brumimicrobium*, as they are structural parts in the co-occurrence network.

The functional potential of the five SH communities showed similar patterns in general, with particular differences in some categories following the same association pattern described before (Fig. 3). Comparing each site, we found significant differences in the enrichment of some important functional categories related to the maintenance of critical geochemical cycles, including carbon, nitrogen, and phosphate metabolism, stress resistance, transport of zinc, nickel, cobalt, iron, and manganese, and osmotic stress. Notably, the two functions with statistical differences between almost all communities were the circadian clock in *Cyanobacteria* and the Calvin Benson cycle, suggesting differences in primary production and $CO_2$ fixation. The circadian rhythm is directly influenced by several abiotic conditions, such as temperature and atmospheric pressure (51). Moreover, the Calvin Benson cycle was the only metabolic pathway involved in $CO_2$ fixation reported in an endolithic halite metagenome analysis, underlining its importance for microbial communities living under these extreme conditions (52). This agrees with the results of the MEBS analysis, where the relevant reactions of the carbon cycle presented a greater variation between the metagenomes. Particularly, carbon flux control in these communities seems to be different, especially regarding methane processes, as the far greater detection of specific markers in H0 and H1 could account for the high presence of methanogen archaea in these two communities (53). Previous studies have reported differences in the chemical compositions for these two sites, including lower concentrations of $Ca^{2+}$, $Mg^{2+}$, $K^+$, $Na^+$, ammonia, nitrates, phosphates, sulfates, and other nutrients (5, 6, 54), therefore making it more appropriate for bacteria with different metabolisms. This could also be related with the low S in H1, implying that methanogens outcompete sulfate reducers in this community (55).

Furthermore, among these significant differences, the As resistance category is enriched in the metagenomes from the sites with the highest As concentration, as expected and reported previously (56, 57). Particularly, the distribution and abundance of As-related genes in the five SH communities showed significant differences between the communities as well (Fig. 4). Therefore, As could be among the environmental parameters that influence the selection and dynamics of these communities (58, 59). Our results agree with this premise, since the abundance of As-related genes increases as As levels rise between the five sites, indicating a different dynamic in each niche, even though they are geographically close (<6 km).

Overall, we found that most of the detected genes related to As belong to expulsion and reduction mechanisms (*arsR*, *arsC*, and *acr3*), which correlates with the findings reported in the Argentinian Altiplano (19). The transcriptional repressor ArsR was the most abundant. This protein participates in the regulation of different As pumps, namely, ArsB, Acr3, and ArsP (60, 61), and has also been reported to be in operon conformations along with ArsC or ArsM (62). Hence, it was expected to be the most abundant and broadly distributed. Moreover, to better assess the proportion of the As mechanisms found in SH, we separated the genes related to As expulsion out of the cell from those related to reduction (usually reported together). This is due to the fact that gene clusters, including *arsB*, *arsP*, *arsK*, *arsJ*, and *acr3*, are commonly found/

reported independent of arsenate reductase *arsC* (8, 61, 63, 64). Also, arsenic methyltransferase (ArsM) was the next most abundant protein, and particularly enriched in H3, H4, and H5, suggesting that this protein is part of a complementary mechanism enabling hosts to thrive in sites with high As concentrations (65).

Even though proteins related to oxidation and respiration of As were less abundant, their distribution is interesting, which could be due to niche-specific conditions. Although these mechanisms are somewhat known and recurrent, much remains unclear, mostly regarding their function and interconnection with the central metabolism of bacteria (66, 67). This problem is exacerbated when working in unexplored environments, where unknown or unclassified organisms abound. Besides, the amount of misclassified and/or unclassified protein sequences in databases is a major limitation for these investigations. Nonetheless, these difficulties have been fully acknowledged and are being addressed (68–71). For example, our group was able to manually identify the ArsP and ArsK transporters as well as a possible As respiration system and a missing As methylase in *Exiguobacterium* genomes by using a combination of bioinformatic tools (8). Nevertheless, database shortages and problems caused by classification by homology must always be considered when gene or metabolism detection is being carried out.

Furthermore, we found that the organoarsenical permease *arsP* was detected in the sites with higher As concentrations, which may be a measure to counteract a possibly greater production of these highly toxic organic species; concurrently, the arsenite methyltransferase (*arsM*) gene was also consistently enriched. Thus, these markers could be proposed as bioindicators of As contamination, such as *acr3* (72). Additionally, the distribution and phylogenetic relationships of As pumps among the metagenomes revealed distinct groups (Fig. 5), consistent with those previously reported: (I) ArsB-Acr3 (ion/BART), (II) ArsP (permease), and (III) ArsK-ArsJ (MFS) (64). It has been hypothesized that the origin of these different mechanisms was due to geological changes of the Earth, to adapt for a particular function and to face emerging As species (73).

The 12 reconstructed high-quality MAGs provide information about the abundance and distribution of undescribed or unknown microorganisms (Fig. 6) as well as genomic insights into widespread As metabolism. Most of these are affiliated within *Proteobacteria* and *Bacteroidetes* phyla, reflecting what we described at the community level. As stated before, the recovery of MAGs complements decades of cultivation and PCR survey efforts, by providing information about taxa missing in culture collections (candidate phyla radiation), improving our understanding of microbial communities (74), particularly those inhabiting extreme environments. Nonetheless, four MAGs were classified at the genus rank: *Arthrospira*, a *Cyanobacteria* with great commercial interest as a pigment source and to make spirulina supplements; *Thioalkalivibrio*, an aerobic sulfur oxidizer; and *Desulfatitalea* and *Desulfosarcina*, two anaerobic sulfate reducers (75–77).

In the same way that these MAGs only cover the most abundant phyla of the SH, we only detected genes that belong to the most prevalent As resistance mechanisms in these MAGs (Fig. 7), i.e., expulsion, reduction, and methylation mechanisms. Conversely, no As oxidation- and/or respiration-related genes were detected in the recovered MAGs. This indicates that the different ways used to expel this metalloid from the cell could be more direct, constituting a first line of defense against As, while oxidation and respiration would involve coupling with the central metabolism of the bacteria and a more complex machinery to benefit from this compound, which seems to be associated with less abundant and highly adapted or specialized bacteria. Also, As oxidation and respiration have been reported in association with nitrate and sulfate reduction, respectively (9, 78). Nonetheless, we need to take into consideration the completion levels of the MAGs as a possible source for missing genes.

Overall, our results reveal that populations of *Proteobacteria*, *Bacteroidetes*, and *Cyanobacteria* are abundant across broad ecological niches in the SH, spanning a challenging ensemble of environmental conditions and physicochemical parameters, of which As is highly relevant. As the As cycle and the contribution played by bacteria in it have yet to be completely understood, more environmental studies are needed.

Metagenomic approaches have been shedding light on the possible role of unknown or undescribed microorganisms and, combined with transcriptomic, metabolomic, and cultivation strategies, will be essential to define these phenomena and their relevance in a global context.

**Conclusion.** The Altiplano array of ecological niches is a reservoir for microbial diversity, showing great richness in adapted organisms capable of facing these challenging conditions. Particularly, the Salar de Huasco is a highly diverse ecosystem, where salinity and As concentrations contribute to shaping the community composition, mainly represented by *Proteobacteria* and *Bacteroidetes*. Also, the interaction networks within these communities show three distinct groups of related taxa, but no keystone nodes were found. Nonetheless, smaller niche overlaps were determined. Altogether, these indicate that the niches studied in the SH harbor highly diverse communities, H1 and H5 being the most contrasting ones. Moreover, the most abundant As-related genes found in these communities indicate that the As(V) reduction and subsequent As(III) expulsion systems are the most common strategy used to detoxify the cell of As. Furthermore, regarding expulsion pumps, the most abundant was Acr3, followed by ArsB; however, in sites with high As concentrations, ArsP becomes enriched. In addition, 12 high-quality nonredundant MAGs were reconstructed from the metagenomes. These represented the dominant diversity detected across the communities as well as the metabolic variability and the presence of marker genes related to the most recurrent As resistance mechanisms (expulsion, reduction, and methylation). Finally, in order to further elucidate the strategies and relationships between the microbial taxa and among biotic and abiotic components within the ecosystem, further multidisciplinary studies are required as well as the use of the ever-evolving next-generation sequencing (NGS) approaches. Together with more precise database information, we will be able to better understand the evolutionary process of adaptation to the extreme conditions present in these unique ecosystems.

## MATERIALS AND METHODS

**Study area and sampling.** The Salar de Huasco National Park is an area located on the Chilean Altiplano that is known for its spatial heterogeneity, great biodiversity, and physicochemical characteristics. This ecosystem is mostly composed of streams, salt crusts, peatlands, and shallow (permanent and nonpermanent) lakes with salinity and As gradients from north to south (8). In June 2018, we sampled sediments (by duplicate and to a depth of 5 cm) from five different sites: the main water body (H3, H4, and H5) and the stream that feeds it (H0 and H1) approximately 1 m inward from the shore line. Samples were collected in sterile tubes and kept and transported in a cooler until storage at −20°C for subsequent DNA extraction. Physicochemical parameters, including temperature, salinity, and pH, were recorded (HI 98192 and HI 2211, HANNA Instruments) *in situ* (Fig. 1 and see Table S1 in the supplemental material).

**DNA extraction and high-throughput shotgun sequencing.** Total DNA was extracted from the sediment samples from each SH site using the DNeasy PowerSoil kit (Qiagen Inc., Hilden, Germany) according to the manufacturer's instructions. DNA integrity, quality, and quantity were verified through 1% agarose gel electrophoresis, optical density at 260 nm to optical density at 280 nm ($OD_{260/280}$) ratio, and fluorescence using a Qubit 3.0 fluorometer along with the Qubit dsDNA HS assay kit (Thermo Fisher Scientific, MA, USA). Next, paired-end (150 bp) libraries were constructed for each sample in duplicates at the Centro de Biotecnología Vegetal, Universidad Andrés Bello (Santiago, Chile) using the TruSeq Nano DNA kit (Illumina Inc., CA, USA) according to the TruSeq Nano DNA sample preparation guide 15041110 revision D. Libraries were sent for sequencing at Macrogen Inc. (Seoul, South Korea) on a HiSeq 4000 platform (Illumina Inc.). Then, raw data were evaluated using FastQC v0.11.8 (79) for quality control, and adapters were removed from the reads of all samples using Trimmomatic v0.30 (80) and then filtered and trimmed (length ≤ 100 bp, Ns = 0, and Q ≤ 30 thresholds) with PRINSEQ v0.20.4 (81).

**Taxonomic profiling analysis.** Quality-controlled reads for each sample were profiled using the phyloFlash pipeline (82) to obtain all reads that align with the bacterial small-subunit rRNA (SSU rRNA) SILVA v132 database (83). Subsequently, these sequences (FASTQ files) were processed using R v3.5.2 and RStudio v1.1.463 (84, 85) following the DADA2 v1.16.0 R package pipeline (86) to infer amplicon sequence variants (ASVs) present in each sample. Briefly, after dereplication, denoising, and paired-reads merge steps, the ASV table was built with 97% clustering, the chimeras were removed, and the taxonomic assignments were made using the SILVA v132 database (83) using DADA2 Ribosomal Database Project's (RDP) naive Bayesian classifier (87). Then, data were normalized by variance stabilizing transformation using the R package DESeq2 v1.28.1 (88). Also, a multiple-sequence alignment was created using the R package DECIPHER v2.16.1 (89) to infer a phylogeny with FastTree v2.1.10 (90). Furthermore, a phyloseq object (containing the ASVs, taxonomy assignation, phylogenetic tree, and sample meta-data) was created using the R package Phyloseq v1.32.0 (91) in order to calculate the alpha diversity indexes, along

with btools v0.0.1 R package. Also, beta diversity (principal-coordinate analysis [PCoA] Bray Curtis distance with environmental variables fit) was calculated using the R package ampvis2 v2.4.5 (92), and visualizations were generated with the ggplot2 v2.2.1 (93) R package.

**Co-occurrence networks.** We used the same phyloseq object, which was agglomerated by best hit using the microbiomeutilities v1.00.11 R package (94), and then filtered by tax abundance (0.5% in at least one sample) using Genefilter v 1.72.0 (95) and Phyloseq v1.32.0 (91) R packages. Then, the co-occurrence network was estimated using the SPIEC-EASI (SParse InversE Covariance Estimation for Ecological Association Inference) v0.1.4 (96) R package, using the neighborhood selection model (parameters: lambda.min.ratio = 1e−2, nlambda = 20, and 50 replicates). Finally, the network was visualized using the ggnet2 function of the GGally v1.5.0 (97) R package (a ggplot2 extension).

**Functional profiling analysis.** The patterns of functional potential as subsystems with different specificity levels for each community were determined by the presence/absence and relative abundance of the quality-controlled reads that aligned against the metabolic pathways and functions in the SEED database (98) with SUPER-FOCUS v0.35 software (99), which uses DIAMOND v2.0.6 (100) for fast and efficient alignment. Statistical analysis was carried out in STAMP (Software package for Analyzing Taxonomic or Metabolic Profiles) v2.3.1 (101) using Welch's $t$ test to compare all samples.

**Metagenome coassembly and read mapping.** As we needed for subsequent analyses (i) a coassembly (the reads from all samples assembled together [.fasta file]), (ii) its annotations (.gff, .ffn, and .faa files), and (iii) an alignment per sample (all the reads from each sample mapped against the coassembly [.bam files]), we proceeded to assemble the quality-controlled reads from all samples using MEGAHIT v1.1.3 (102), with the presets meta-large option and a minimum contig length of 1 kb. The coassembly was then evaluated with MetaQUAST v5.0.2 (103) and annotated with Prokka v1.11 (104) using the metagenome mode. Furthermore, we mapped the quality-controlled reads from each sample against the coassembly using Bowtie2 v2.3.4 (105) and stored the recruited reads (sorted and indexed) as BAM files using SAMtools v1.3 (106).

**Metagenome profiling.** We followed the anvi'o v7 pipeline (107). First, we created a contig database with the coassembled contigs, which uses Prodigal v2.6.3 (108), HMMER v3.3.1 (109), and NCBI Clusters of Orthologous Genes (COGs) (110) to identify gene calls and functionally annotate them (anvi-gen-contigs-database, anvi-run-hmms, anvi-run-ncbi-cogs). Second, we profiled each sample's BAM file against the contigs database to estimate the detection and coverage statistics for each contig (anvi-profile). Then, we combined the profiles in a single merged metagenomic profile database, which uses all individual statistics to compute hierarchical clustering of the contigs (anvi-merge). Finally, we visualized the merged profile on the anvi'o interactive interface, which allows easy exploration and curation of the metagenomes (anvi-interactive). On the other hand, we used the MEBS (multigenomic entropy-based score) v1.0 package (111) to evaluate and compare the S, N, O, $CH_4$, and Fe biogeochemical cycles in each metagenome using the predicted proteins (annotated .faa file from each individual assembly).

**Metagenome target gene search.** To estimate the abundance of genes related to As metabolism, the read counts for each predicted gene of each sample were obtained using the corresponding BAM file and the coassembly annotated GFF file, with HTSeq-Counts v0.13.5 (112). Also, we constructed a database from the GenBank "Identical Protein Groups" with all available prokaryotic proteins of As metabolism (Acr3, AioA, AioB, AioR, AioS, AioX, AoxA, AoxB, AoxC, AoxD, ArrA, ArrB, ArrC, ArsA, ArsB, ArsC, ArsD, ArsH, ArsJ, ArsK, ArsM, ArsN, ArsO, ArsP, ArsR, ArsT, ArxA, ArxB, ArxR, ArxS, and ArxX). This was then targeted with the coassembly predicted proteins (.faa file) using CRB-BLAST v0.6.6 (113) (E value ≥ 1E−05, identity ≥ 70%, and query coverage ≥ 70%). Finally, matching the hits with the gene counts (normalized by the target gene length and the corresponding library size), we calculated the relative abundance of the genes of interest for the five metagenomes. Visualizations were made with R packages ggplot2 v2.2.1 (93) and pheatmap v1.0.12 (114). Furthermore, all the detected sequences that corresponded to As efflux pumps were extracted from the .ffn file and aligned using MAFFT v7 (115). Arsenic phylogeny inference was calculated with FastTree v2.1.10 (90) and visualized with the anvi'o v7 (107) interactive interface.

**Metagenomic binning.** The contigs in the metagenome profile were clustered through anvi'o v7 (107) using the CONCOCT v1.1.0 (116) binning program, which adds a bins collection to the profile (anvi-cluster-contigs). The resulting bins were evaluated for completion/redundancy, and manual curation/refinement was carried out in the interactive interface (anvi-estimate-genome-completeness and anvi-refine). Finally, the refined bins were displayed in the interactive interface and summarized to obtain all the statistics and files for downstream analysis (anvi-interactive and anvi-summarize).

**MAGs evaluation, taxonomy, and functional estimation.** We define MAGs (metagenome assembles genomes) as bins with a completion of >80% and redundancy of <10%. Then, we used CheckM v1.0.13 (117) for a more robust evaluation. Subsequently, to infer MAG taxonomy, we used GTDB-Tk v0.3.2 (118) along with the Genome Taxonomy Database (119). Moreover, we estimated the MAGs' metabolic potential by evaluating their gene content with anvi'o v7 (107). To do so, first, functions and metabolic pathways were annotated to the MAGs using HMM hits from the KOfam KEGG Orthologs (KO) database (120, 121) (anvi-run-kegg-kofams). Second, relying on these KO annotations, the metabolic pathways were predicted considering those defined by KOs in the KEGG MODULES resource (122), where a KO represents a gene function, and a module represents a group of KOs that together carry out the reactions in a metabolic path (anvi-estimate-metabolism). The MAGs module completion was visualized using pheatmap v1.0.12 R package (114).

**MAGs target gene search.** All MAGs were queried against the same previously constructed database (with the As metabolism genes) using CRB-BLAST v0.6.6 (113) (E value ≥ 1E−05, identity ≥ 70%, and query coverage ≥ 70%). Furthermore, the resulting hits matrix was compared and supplemented

with the detected KOfams related to As, and the final matrix was used to generate a functional network using Gephi v0.9.2 (123) to connect MAGs and detected genes.

**Data availability.** The whole raw data sets and the metagenome assembled genomes have been deposited at DDBJ/ENA/GenBank under the BioProject accession number PRJNA573913.

## SUPPLEMENTAL MATERIAL

Supplemental material is available online only.

**SUPPLEMENTAL FILE 1**, PDF file, 0.6 MB.
**SUPPLEMENTAL FILE 2**, TXT file, 0.3 MB.
**SUPPLEMENTAL FILE 3**, TXT file, 0.01 MB.
**SUPPLEMENTAL FILE 4**, TXT file, 0.05 MB.

## ACKNOWLEDGMENTS

We thank the illustrator Florence Gutzwiller (IG: @spideryscrawl) for the SH landscape painting (https://spideryscrawl-illustration.webnode.com/). We thank Universidad Andres Bello's high-performance computing cluster, Dylan (http://www.castrolab.org/), for providing data storage, support, and computing power for bioinformatic analyses. We also thank the MerenLab group (https://merenlab.org/) for providing help, advice, and solutions related to the use of Anvi'o. We thank Valerie de Anda (https://valdeanda.github.io/) as well, for guidance and advice in implementing the MEBS software. Finally, we thank Michael G. Handford for the technical English writing edition.

J.C.-S., C.P.-E., E.C.-N., F.R., and C.P.S. conceived and designed the study. J.C.-S., J.F., F.M., F.R., and C.P.S. performed the field work. J.C.-S. and C.P.-E. processed the samples, performed the experimental procedures and carried out the bioinformatics analyses. K.N.M., S.M., and E.C.-N. contributed with critical bioinformatics advice. C.P.S., F.R., and E.C.-N. contributed with reagents, materials, and analysis tools. J.C.-S. and C.P.-E. interpreted the results and wrote the first manuscript draft. All authors read and approved the final manuscript.

This research was sponsored by ANID (Agencia Nacional de Investigación y Desarrollo de Chile) grants. C.P.S. was funded by ANID-FONDECYT regular 1210633 and ECOS-ANID 170023. E.C.-N. was funded by ANID-FONDECYT regular 1200834 and the US Air Force Office for Scientific Research (FA9550-20-1-0337). J.C.-S. was funded by ANID 2021 postdoctoral FONDECYT 3210156. F.R. was funded by Chilean Ministry of Education (MINEDUC) PMI 1795 Project. J.F. was funded by Universidad Católica del Norte 2020 Doctoral Scholarship.

The funders had no role in study design, data collection and analysis, decision to publish, or preparation of the manuscript. We declare that the research was conducted in the absence of any commercial or financial relationships that could be construed as a potential conflict of interest.

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
