## [Reviewer comments · Microbiology Spectrum]

Microbiology Spectrum

Living to the high extreme: unraveling the composition, structure, and functional insights of bacterial communities thriving in the arsenic-rich Salar de Huasco – Altiplanic ecosystem.

Juan Castro-Severyn, Coral Pardo-Esté, Katterinne Mendez, Jonathan Fortt, Sebastián Márquez, Franck Molina, Eduardo Castro-Nallar, Francisco Remonsellez, and Claudia Saavedra

Corresponding Author(s): Claudia Saavedra, Universidad Andres Bello

Review Timeline:

Submission Date:

June 2, 2021

Accepted:

June 7, 2021

Editor: Jeffrey Gralnick

Reviewer(s): The reviewers have opted to remain anonymous.

Transaction Report:

DOI: <https://doi.org/10.1128/Spectrum.00444-21>

June 7, 2021

Dr. Claudia P. Saavedra
Universidad Andres Bello
Lab. Microbiología Molecular, Ciencias de la Vida
Republica 330
Universidad Católica del Norte
Antofagasta, Santiago 1270709
Chile

Re: Spectrum00444-21 (Living to the high extreme: unraveling the composition, structure, and functional insights of bacterial communities thriving in the arsenic-rich Salar de Huasco - Altiplanic ecosystem.)

Dear Dr. Claudia P. Saavedra:

Your manuscript has been accepted, and I am forwarding it to the ASM Journals Department for publication. You will be notified when your proofs are ready to be viewed. I really enjoyed reading your manuscript and I know it will be of great interest to researchers who think about extreme environments and arsenic metabolism.

Sincerely,

Jeffrey Gralnick
Editor, Microbiology Spectrum

Journals Department
Table S5: Accept

Table S7: Accept

Supplemental_Material_FOR_Publication: Accept

Table S3: Accept